# Online Rank Elicitation for Plackett-Luce:
# A Dueling Bandits Approach

**Balázs Szörényi**
Technion, Haifa, Israel /
MTA-SZTE Research Group on
Artificial Intelligence, Hungary
szorenyibalazs@gmail.com

**Róbert Busa-Fekete, Adil Paul, Eyke Hüllermeier**
Department of Computer Science
University of Paderborn
Paderborn, Germany
{busarobi,adil.paul,eyke}@upb.de

## Abstract

We study the problem of online rank elicitation, assuming that rankings of a set of alternatives obey the Plackett-Luce distribution. Following the setting of the dueling bandits problem, the learner is allowed to query pairwise comparisons between alternatives, i.e., to sample pairwise marginals of the distribution in an online fashion. Using this information, the learner seeks to reliably predict the most probable ranking (or top-alternative). Our approach is based on constructing a surrogate probability distribution over rankings based on a sorting procedure, for which the pairwise marginals provably coincide with the marginals of the Plackett-Luce distribution. In addition to a formal performance and complexity analysis, we present first experimental studies.

## 1 Introduction

Several variants of learning-to-rank problems have recently been studied in an online setting, with preferences over alternatives given in the form of stochastic pairwise comparisons [6]. Typically, the learner is allowed to select (presumably most informative) alternatives in an active way—making a connection to multi-armed bandits, where single alternatives are chosen instead of pairs, this is also referred to as the *dueling bandits* problem [28].

Methods for online ranking can mainly be distinguished with regard to the assumptions they make about the probabilities $p_{i,j}$ that, in a direct comparison between two alternatives $i$ and $j$, the former is preferred over the latter. If these probabilities are not constrained at all, a complexity that grows quadratically in the number $M$ of alternatives is essentially unavoidable [27, 8, 9]. Yet, by exploiting (stochastic) transitivity properties, which are quite natural in a ranking context, it is possible to devise algorithms with better performance guaranties, typically of the order $M \log M$ [29, 28, 7].

The idea of exploiting transitivity in preference-based online learning establishes a natural connection to *sorting algorithms*. Naively, for example, one could simply apply an efficient sorting algorithm such as MergeSort as an active sampling scheme, thereby producing a random order of the alternatives. What can we say about the optimality of such an order? The problem is that the probability distribution (on rankings) induced by the sorting algorithm may not be well attuned with the original preference relation (i.e., the probabilities $p_{i,j}$).

In this paper, we will therefore combine a sorting algorithm, namely QuickSort [15], and a stochastic preference model that harmonize well with each other—in a technical sense to be detailed later on. This harmony was first presented in [1], and our main contribution is to show how it can be exploited for online rank elicitation. More specifically, we assume that pairwise comparisons obey the marginals of a Plackett-Luce model [24, 19], a widely used parametric distribution over rankings (cf. Section 5). Despite the quadratic worst case complexity of QuickSort, we succeed in developing its budgeted version (presented in Section 6) with a complexity of $\mathcal{O}(M \log M)$. While only returning partial orderings, this version allows us to devise PAC-style algorithms that find, respectively, a close-to-optimal item (Section 7) and a close-to-optimal ranking of all items (Section 8), both with high probability.

## 2 Related Work

Several studies have recently focused on preference-based versions of the multi-armed bandit setup, also known as dueling bandits [28, 6, 30], where the online learner is only able to compare arms in a pairwise manner. The outcome of the pairwise comparisons essentially informs the learner about pairwise preferences, i.e., whether or not an option is preferred to another one. A first group of papers, including [28, 29], assumes the probability distributions of pairwise comparisons to possess certain regularity property, such as strong stochastic transitivity. A second group does not make assumptions of that kind; instead, a target ("ground-truth") ranking is derived from the pairwise preferences, for example using the Copeland, Borda count and Random Walk procedures [9, 8, 27]. Our work is obviously closer to the first group of methods. In particular, the study presented in this paper is related to [7] which investigates a similar setup for the Mallows model.

There are several approaches to estimating the parameters of the Plackett-Luce (PL) model, including standard statistical methods such as likelihood estimation [17] and Bayesian parameter estimation [14]. Pairwise marginals are also used in [26], in connection with the method-of-moments approach; nevertheless, the authors assume that *full* rankings are observed from a PL model.

Algorithms for *noisy sorting* [2, 3, 12] assume a total order over the items, and that the comparisons are representative of that order (if $i$ precedes $j$, then the probability of option $i$ being preferred to $j$ is bigger than some $\lambda > 1/2$). In [25], the data is assumed to consist of pairwise comparisons generated by a Bradley-Terry model, however, comparisons are not chosen actively but according to some fixed probability distribution.

Pure exploration algorithms for the stochastic multi-armed bandit problem sample the arms a certain number of times (not necessarily known in advance), and then output a recommendation, such as the best arm or the $m$ best arms [4, 11, 5, 13]. While our algorithms can be viewed as pure exploration strategies, too, we do not assume that *numerical* feedback can be generated for *individual* options; instead, our feedback is *qualitative* and refers to *pairs* of options.

## 3 Notation

A set of alternatives/options/items to be ranked is denoted by $\mathcal{I}$. To keep the presentation simple, we assume that items are identified by natural numbers, so $\mathcal{I} = [M] = \{1, \ldots, M\}$. A *ranking* is a bijection $\mathbf{r}$ on $\mathcal{I}$, which can also be represented as a vector $\mathbf{r} = (r_1, \ldots, r_M) = (\mathbf{r}(1), \ldots, \mathbf{r}(M))$, where $r_j = \mathbf{r}(j)$ is the rank of the $j$th item. The set of rankings can be identified with the symmetric group $\mathbb{S}_M$ of order $M$. Each ranking $\mathbf{r}$ naturally defines an associated *ordering* $\mathbf{o} = (o_1, \ldots, o_M) \in \mathbb{S}_M$ of the items, namely the inverse $\mathbf{o} = \mathbf{r}^{-1}$ defined by $o_{\mathbf{r}(j)} = j$ for all $j \in [M]$.

For a permutation $\mathbf{r}$, we write $\mathbf{r}(i, j)$ for the permutation in which $r_i$ and $r_j$, the ranks of items $i$ and $j$, are replaced with each other. We denote by $\mathcal{L}(r_i = j) = \{\mathbf{r} \in \mathbb{S}_M \,|\, r_i = j\}$ the subset of permutations for which the rank of item $i$ is $j$, and by $\mathcal{L}(r_j > r_i) = \{\mathbf{r} \in \mathbb{S}_M \,|\, r_j > r_i\}$ those for which the rank of $j$ is higher than the rank of $i$, that is, item $i$ is preferred to $j$, written $i \succ j$. We write $i \succ_{\mathbf{r}} j$ to indicate that $i$ is preferred to $j$ with respect to ranking $\mathbf{r}$.

We assume $\mathbb{S}_M$ to be equipped with a probability distribution $\mathbb{P} : \mathbb{S}_M \to [0, 1]$; thus, for each ranking $\mathbf{r}$, we denote by $\mathbb{P}(\mathbf{r})$ the probability to observe this ranking. Moreover, for each pair of items $i$ and $j$, we denote by

$$p_{i,j} = \mathbb{P}(i \succ j) = \sum_{\mathbf{r} \in \mathcal{L}(r_j > r_i)} \mathbb{P}(\mathbf{r}) \tag{1}$$

the probability that $i$ is preferred to $j$ (in a ranking randomly drawn according to $\mathbb{P}$). These pairwise probabilities are called the *pairwise marginals* of the ranking distribution $\mathbb{P}$. We denote the matrix composed of the values $p_{i,j}$ by $\mathbf{P} = [p_{i,j}]_{1 \le i,j \le M}$.

## 4 Preference-based Approximations

Our learning problem essentially consists of making good predictions about properties of $\mathbb{P}$. Concretely, we consider two different goals of the learner, depending on whether the application calls for the prediction of a single item or a full ranking of items:

In the first problem, which we call **PAC-Item** or simply **PACI**, the goal is to find an item that is almost as good as the optimal one, with optimality referring to the Condorcet winner. An item $i^*$ is

a Condorcet winner if $p_{i^*,i} > 1/2$ for all $i \neq i^*$. Then, we call an item $j$ a PAC-item, if it is beaten by the Condorcet winner with at most an $\epsilon$-margin: $|p_{i^*,j} - 1/2| < \epsilon$. This setting coincides with those considered in [29, 28]. Obviously, it requires the existence of a Condorcet winner, which is indeed guaranteed in our approach, thanks to the assumption of a Plackett-Luce model.

The second problem, called **AMPR**, is defined as finding the most probable ranking [7], that is, $\mathbf{r}^* = \operatorname{argmax}_{\mathbf{r} \in \mathbb{S}_M} \mathbb{P}(\mathbf{r})$. This problem is especially challenging for ranking distributions for which the order of two items is hard to elicit (because many entries of $\mathbf{P}$ are close to $1/2$). Therefore, we again relax the goal of the learner and only require it to find a ranking $\mathbf{r}$ with the following property: There is no pair of items $1 \leq i, j \leq M$, such that $r_i^* < r_j^*$, $r_i > r_j$ and $p_{i,j} > 1/2 + \epsilon$. Put in words, the ranking $\mathbf{r}$ is allowed to differ from $\mathbf{r}^*$ only for those items whose pairwise probabilities are close to $1/2$. Any ranking $\mathbf{r}$ satisfying this property is called an approximately most probable ranking (AMPR).

Both goals are meant to be achieved with probability at least $1 - \delta$, for some $\delta > 0$. Our learner operates in an online setting. In each iteration, it is allowed to gather information by asking for a single *pairwise comparison* between two items—or, using the dueling bandits jargon, to pull two arms. Thus, it selects two items $i$ and $j$, and then observes either preference $i \succ j$ or $j \succ i$; the former occurs with probability $p_{i,j}$ as defined in (1), the latter with probability $p_{j,i} = 1 - p_{i,j}$. Based on this observation, the learner updates its estimates and decides either to continue the learning process or to terminate and return its prediction. What we are mainly interested in is the sample complexity of the learner, that is, the number of pairwise comparisons it queries prior to termination.

Before tackling the problems introduced above, we need some additional notation. The pair of items chosen by the learner in the $t$-th comparison is denoted $(i^t, j^t)$, where $i^t < j^t$, and the feedback received is defined as $o^t = 1$ if $i^t \succ j^t$ and $o^t = 0$ if $j^t \succ i^t$. The set of steps among the first $t$ iterations in which the learner decides to compare items $i$ and $j$ is denoted by $I_{i,j}^t = \{\ell \in [t] \,|\, (i^\ell, j^\ell) = (i,j)\}$, and the size of this set by $n_{i,j}^t = \#I_{i,j}^t$.[1] The proportion of "wins" of item $i$ against item $j$ up to iteration $t$ is then given by $\widehat{p}_{i,j}^t = \frac{1}{n_{i,j}^t} \sum_{\ell \in I_{i,j}^t} o^\ell$. Since our samples are independent and identically distributed (i.i.d.), the relative frequency $\widehat{p}_{i,j}^t$ is a reasonable estimate of the pairwise probability (1).

## 5   The Plackett-Luce Model

The Plackett-Luce (PL) model is a widely-used probability distribution on rankings [24, 19]. It is parameterized by a "skill" vector $\mathbf{v} = (v_1, \ldots, v_M) \in \mathbb{R}_+^M$ and mimics the successive construction of a ranking by selecting items position by position, each time choosing one of the remaining items $i$ with a probability proportional to its skill $v_i$. Thus, with $\mathbf{o} = \mathbf{r}^{-1}$, the probability of a ranking $\mathbf{r}$ is

$$\mathbb{P}(\mathbf{r} \,|\, \mathbf{v}) = \prod_{i=1}^{M} \frac{v_{o_i}}{v_{o_i} + v_{o_{i+1}} + \cdots + v_{o_M}} \quad . \tag{2}$$

As an appealing property of the PL model, we note that the marginal probabilities (1) are very easy to calculate [21], as they are simply given by

$$p_{i,j} = \frac{v_i}{v_i + v_j} \quad . \tag{3}$$

Likewise, the most probable ranking $\mathbf{r}^*$ can be obtained quite easily, simply by sorting the items according to their skill parameters, that is, $r_i^* < r_j^*$ iff $v_i > v_j$. Moreover, the PL model satisfies strong stochastic transitivity, i.e., $p_{i,k} \geq \max(p_{i,j}, p_{j,k})$ whenever $p_{i,j} \geq 1/2$ and $p_{j,k} \geq 1/2$ [18].

## 6   Ranking Distributions based on Sorting

In the classical sorting literature, the outcome of pairwise comparisons is deterministic and determined by an underlying total order of the items, namely the order the sorting algorithm seeks to find. Now, if the pairwise comparisons are stochastic, the sorting algorithm can still be run, however, the result it will return is a random ranking. Interestingly, this is another way to define a probability distribution over the rankings: $\mathbb{P}(\mathbf{r}) = \mathbb{P}(\mathbf{r} \,|\, \mathbf{P})$ is the probability that $\mathbf{r}$ is returned by the algorithm if

stochastic comparisons are specified by $\mathbf{P}$. Obviously, this view is closely connected to the problem of noisy sorting (see the related work section).

In a recent work by Ailon [1], the well-known QuickSort algorithm is investigated in a stochastic setting, where the pairwise comparisons are drawn from the pairwise marginals of the Plackett-Luce model. Several interesting properties are shown about the ranking distribution based on QuickSort, notably the property of *pairwise stability*. We denote the QuickSort-based ranking distribution by $\mathbb{P}_{QS}(\cdot \,|\, \mathbf{P})$, where the matrix $\mathbf{P}$ contains the marginals (3) of the Plackett-Luce model. Then, it can be shown that $\mathbb{P}_{QS}(\cdot \,|\, \mathbf{P})$ obeys the property of pairwise stability, which means that it preserves the marginals, although the distributions themselves might not be identical, i.e., $\mathbb{P}_{QS}(\cdot \,|\, \mathbf{P}) \neq \mathbb{P}(\cdot \,|\, \mathbf{v})$.

**Theorem 1** (Theorem 4.1 in [1]). *Let $\mathbf{P}$ be given by the pairwise marginals (3), i.e., $p_{i,j} = v_i/(v_i + v_j)$. Then, $p_{i,j} = \mathbb{P}_{QS}(i \succ j \,|\, \mathbf{P}) = \sum_{\mathbf{r} \in \mathcal{L}(r_j > r_i)} \mathbb{P}_{QS}(\mathbf{r} \,|\, \mathbf{P})$.*

One drawback of the QuickSort algorithm is its complexity: To generate a random ranking, it compares $\mathcal{O}(M^2)$ items in the worst case. Next, we shall introduce a budgeted version of the Quick-Sort algorithm, which terminates if the algorithm compares too many pairs, namely, more than $\mathcal{O}(M \log M)$. Upon termination, the modified Quicksort algorithm only returns a partial order. Nevertheless, we will show that it still preserves the pairwise stability property.

## 6.1 The Budgeted QuickSort-based Algorithm

Algorithm 1 shows a budgeted version of the QuickSort-based random ranking generation process described in the previous section. It works in a way quite similar to the standard QuickSort-based algorithm, with the notable difference of terminating as soon as the number of pairwise comparisons exceeds the budget $B$, which is a parameter assumed as an input. Obviously, the BQS algorithm run with $A = [M]$ and $B = \infty$ (or $B > M^2$) recovers the original QuickSort-based sampling algorithm as a special case.

A run of $\mathrm{BQS}(A, \infty)$ can be represented quite naturally as a random tree $\tau$: the root is labeled $[M]$, end whenever a call to $\mathrm{BQS}(A, B)$ initiates a recursive call $\mathrm{BQS}(A', B')$, a child node with label $A'$ is added to the node with label $A$. Note that each such tree determines a ranking, which is denoted by $\mathbf{r}_\tau$, in a natural way.

---

**Algorithm 1** $\mathrm{BQS}(A, B)$

**Require:** $A$, the set to be sorted, and a budget $B$
**Ensure:** $(\mathbf{r}, B'')$, where $B''$ is the remaining budget, and $\mathbf{r}$ is the (partial) order that was constructed based on $B - B''$ samples
1: Initialize $\mathbf{r}$ to be the empty partial order over $A$
2: **if** $B \leq 0$ or $|A| \leq 1$ **then return** $(\mathbf{r}, 0)$
3: pick an element $i \in A$ uniformly at random
4: **for all** $j \in A \setminus \{i\}$ **do**
5:     draw a random sample $o_{ij}$ according to the PL marginal (3)
6:     update $\mathbf{r}$ accordingly
7: $A_0 = \{j \in A \,|\, j \neq i \ \& \ o_{i,j} = 0\}$
8: $A_1 = \{j \in A \,|\, j \neq i \ \& \ o_{i,j} = 1\}$
9: $(\mathbf{r}', B') = \mathrm{BQS}(A_0, B - |A| + 1)$
10: $(\mathbf{r}'', B'') = \mathrm{BQS}(A_1, B')$
11: update $\mathbf{r}$ based on $\mathbf{r}'$ and $\mathbf{r}''$
12: **return** $(\mathbf{r}, B'')$

---

The random ranking generated by $\mathrm{BQS}(A, \infty)$ for some subset $A \subseteq [M]$ was analyzed by Ailon [1], who showed that it gives back the same marginals as the original Plackett-Luce model (as recalled in Theorem 1). Now, for $B > 0$, denote by $\tau^B$ the tree the algorithm would have returned for the budget $B$ instead of $\infty$. [2] Additionally, let $\mathcal{T}^B$ denote the set of all possible outcomes of $\tau^B$, and for two distinct indices $i$ and $j$, let $\mathcal{T}_{i,j}^B$ denote the set of all trees $T \in \mathcal{T}^B$ in which $i$ and $j$ are incomparable in the associated ranking (i.e., some leaf of $T$ is labelled by a superset of $\{i, j\}$).

The main result of this section is that BQS does not introduce any bias in the marginals (3), i.e., Theorem 1 also holds for the budgeted version of BQS.

**Proposition 2.** *For any $B > 0$, any set $A \subseteq \mathcal{I}$ and any indices $i, j \in A$, the partial order $\mathbf{r} = \mathbf{r}_{\tau^B}$ generated by $\mathrm{BQS}(A, B)$ satisfies $\mathbb{P}(i \succ_\mathbf{r} j \,|\, \tau^B \in \mathcal{T}^B \setminus \mathcal{T}_{i,j}^B) = \frac{v_i}{v_i + v_j}$.*

That is, whenever two items $i$ and $j$ are comparable by the partial ranking $\mathbf{r}$ generated by BQS, $i \succ_\mathbf{r} j$ with probability exactly $\frac{v_i}{v_i + v_j}$. The basic idea of the proof (deferred to the appendix) is to show that, conditioned on the event that $i$ and $j$ are *incomparable* by $\mathbf{r}$, $i \succ_\mathbf{r} j$ would have been

obtained with probability $\frac{v_i}{v_i+v_j}$ in case execution of BQS had been continued (see Claim 6). The result then follows by combining this with Theorem 1.

## 7 The PAC-Item Problem and its Analysis

Our algorithm for finding the PAC item is based on the sorting-based sampling technique described in the previous section. The pseudocode of the algorithm, called PLPAC, is shown in Algorithm 2. In each iteration, we generate a ranking, which is partial (line 6), and translate this ranking into pairwise comparisons that are used to update the estimates of the pairwise marginals. Based on these estimates, we apply a simple elimination strategy, which consists of eliminating an item $i$ if it is significantly beaten by another item $j$, that is, $\widehat{p}_{i,j} + c_{i,j} < 1/2$ (lines 9–11). Finally, the algorithm terminates when it finds a PAC-item for which, by definition, $|p_{i^*,i} - 1/2| < \epsilon$. To identify an item $i$ as a PAC-item, it is enough to guarantee that $i$ is not beaten by any $j \in A$ with a margin bigger than $\epsilon$, that is, $p_{i,j} > 1/2 - \epsilon$ for all $j \in A$. This sufficient condition is implemented in line 12. Since we only have empirical estimates of the $p_{i,j}$ values, the test of the condition does of course also take the confidence intervals into account.

---
**Algorithm 2** PLPAC$(\delta, \epsilon)$
---
1: **for** $i, j = 1 \to M$ **do**  $\quad\quad\quad$ ▷ Initialization
2: $\quad$ $\widehat{p}_{i,j} = 0$ $\quad\quad\quad\quad$ ▷ $\widehat{\mathbf{P}} = [\widehat{p}_{i,j}]_{M \times M}$
3: $\quad$ $n_{i,j} = 0$ $\quad\quad\quad\quad$ ▷ $\widehat{\mathbf{N}} = [n_{i,j}]_{M \times M}$
4: Set $A = \{1, \dots, M\}$
5: **repeat**
6: $\quad$ $\mathbf{r} = \text{BQS}(A, a-1)$ where $a = \#A$ $\quad$ ▷ Sorting based random ranking
7: $\quad$ update the entries of $\widehat{\mathbf{P}}$ and $\mathbf{N}$ corresponding to $A$ based on $\mathbf{r}$
8: $\quad$ set $c_{i,j} = \sqrt{\frac{1}{2n_{i,j}} \log \frac{4M^2 n_{i,j}^2}{\delta}}$ for all $i \neq j$
9: $\quad$ **for** $(i, j \in A) \wedge (i \neq j)$ **do**
10: $\quad\quad$ **if** $\widehat{p}_{i,j} + c_{i,j} < 1/2$ **then**
11: $\quad\quad\quad$ $A = A \setminus \{i\}$ $\quad\quad\quad$ ▷ Discard
12: $\quad$ $C = \{i \in A \mid (\forall j \in A \setminus \{i\})$
$\quad\quad\quad\quad\quad\quad\quad\quad$ $\widehat{p}_{i,j} - c_{i,j} > 1/2 - \epsilon\}$
13: **until** $(\#C \geq 1)$
14: **return** $C$
---

Note that $v_i = v_j$, $i \neq j$, implies $p_{i,j} = 1/2$. In this case, it is not possible to decide whether $p_{i,j}$ is above $1/2$ or not on the basis of a finite number of pairwise comparisons. The $\epsilon$-relaxation of the goal to be achieved provides a convenient way to circumvent this problem.

### 7.1 Sample Complexity Analysis of PLPAC

First, let $\mathbf{r}^t$ denote the (partial) ordering produced by BQS in the $t$-th iteration. Note that each of these (partial) orderings defines a *bucket order*: The indices are partitioned into different classes (buckets) in such a way that none of the pairs are comparable within one class, but pairs from different classes are; thus, if $i$ and $i'$ belong to some class and $j$ and $j'$ belong to some other class, then either $i \succ_{\mathbf{r}^t} j$ and $i' \succ_{\mathbf{r}^t} j'$, or $j \succ_{\mathbf{r}^t} i$ and $j' \succ_{\mathbf{r}^t} i'$. More specifically, the BQS algorithm with budget $a - 1$ (line 6) always results in a bucket order containing only two buckets since no recursive call is carried out with this budget. Then one might show that the optimal arm $i^*$ and an arbitrary arm $i(\neq i^*)$ fall into different buckets "often enough". This observation allows us to upper-bound the number of pairwise comparisons taken by PLPAC with high probability. The proof of the next theorem is deferred to Appendix B.

**Theorem 3.** *Set* $\Delta_i = (1/2)\max\{\epsilon, p_{i^*,i} - 1/2\} = (1/2)\max\{\epsilon, \frac{v_{i^*} - v_i}{2(v_{i^*} + v_i)}\}$ *for each index* $i \neq i^*$.
*With probability at least* $1 - \delta$, *after* $\mathcal{O}\left(\max_{i \neq i^*} \frac{1}{\Delta_i^2} \log \frac{M}{\Delta_i \delta}\right)$ *calls for* BQS *with budget* $M - 1$, PLPAC *terminates and outputs an* $\epsilon$-*optimal arm. Therefore, the total number of samples is* $\mathcal{O}\left(M \max_{i \neq i^*} \frac{1}{\Delta_i^2} \log \frac{M}{\Delta_i \delta}\right)$.

In Theorem 3, the dependence on $M$ is of order $M \log M$. It is easy to show that $\Omega(M \log M)$ is a lower bound, therefore our result is optimal from this point of view.

Our model assumptions based on the PL model imply some regularity properties for the pairwise marginals, such as strong stochastic transitivity and stochastic triangle inequality (see Appendix A of [28] for the proof). Therefore, the INTERLEAVED FILTER [28] and BEAT THE MEAN [29] algorithms can be directly applied in our online framework. Both algorithms achieve a similar sample complexity of order $M \log M$. Yet, our experimental study in Section 9.1 clearly shows that, provided our model assumptions on pairwise marginals are valid, PLPAC outperforms both algorithms in terms of empirical sample complexity.

# 8  The AMPR Problem and its Analysis

For strictly more than two elements, the sorting-based surrogate distribution and the PL distribution are in general not identical, although their mode rankings coincide [1]. The mode $\mathbf{r}^*$ of a PL model is the ranking that sorts the items in decreasing order of their skill values: $r_i < r_j$ iff $v_i > v_j$ for any $i \neq j$. Moreover, since $v_i > v_j$ implies $p_{i,j} > 1/2$, sorting based on the Copeland score $b_i = \#\{1 \leq j \leq M \,|\, (i \neq j) \wedge (p_{i,j} > 1/2)\}$ yields a most probable ranking $\mathbf{r}^*$.

Our algorithm is based on estimating the Copeland score of the items. Its pseudo-code is shown in Algorithm 3 in Appendix C. As a first step, it generates rankings based on sorting, which is used to update the pairwise probability estimates $\widehat{\mathbf{P}}$. Then, it computes a lower and upper bound $\underline{b}_i$ and $\overline{b}_i$ for each of the scores $b_i$. The lower bound is given as $\underline{b}_i = \#\{j \in [M] \setminus \{i\} \,|\, \widehat{p}_{i,j} - c > 1/2\}$, which is the number of items that are beaten by item $i$ based on the current empirical estimates of pairwise marginals. Similarly, the upper bound is given as $\overline{b}_i = \underline{b}_i + s_i$, where $s_i = \#\{j \in [M] \setminus \{i\} \,|\, 1/2 \in [\widehat{p}_{i,j} - c, \widehat{p}_{i,j} + c]\}$. Obviously, $s_i$ is the number of pairs for which, based on the current empirical estimates, it cannot be decided whether $p_{i,j}$ is above or below $1/2$.

As an important observation, note that there is no need to generate a full ranking based on sorting in every case, because if $[\underline{b}_i, \overline{b}_i] \cap [\underline{b}_j, \overline{b}_j] = \emptyset$, then we already know the order of items $i$ and $j$ with respect to $\mathbf{r}^*$. Motivated by this observation, consider the interval graph $G = ([M], E)$ based on the $[\underline{b}_i, \overline{b}_i]$, where $E = \{(i,j) \in [M]^2 \,|\, [\underline{b}_i, \overline{b}_i] \cap [\underline{b}_j, \overline{b}_j] \neq \emptyset\}$. Denote the connected components of this graph by $C_1, \ldots, C_k \subseteq [M]$. Obviously, if two items belong to different components, then they do not need to be compared anymore. Therefore, it is enough to call the sorting-based sampling with the connected components.

Finally, the algorithm terminates if the goal is achieved (line 20). More specifically, it terminates if there is no pair of items $i$ and $j$, for which the ordering with respect to $\mathbf{r}^*$ is not elicited yet, i.e., $[\underline{b}_i, \overline{b}_i] \cap [\underline{b}_j, \overline{b}_j] \neq \emptyset$, and their pairwise probabilities is close to $1/2$, i.e., $|p_{i,j} - 1/2| < \epsilon$.

## 8.1  Sample Complexity Analysis of PLPAC-AMPR

Denote by $q_M$ the expected number of comparisons of the (standard) QuickSort algorithm on $M$ elements, namely, $q_M = 2M \log M + \mathcal{O}(\log M)$ (see e.g., [22]). Thanks to the concentration property of the performance of the QuickSort algorithm, there is no pair of items that falls into the same bucket "too often" in bucket order which is output by BQS. This observation allows us to upper-bound the number of pairwise comparisons taken by PLPAC-AMPR with high probability. The proof of the next theorem is deferred to Appendix D.

**Theorem 4.** *Set $\Delta'_{(i)} = (1/2) \max\{\epsilon, \frac{v_{(i+1)} - v_{(i)}}{2(v_{(i+1)} + v_{(i)})}\}$ for each $1 \leq i \leq M$, where $v_{(i)}$ denotes the $i$-th largest skill parameter. With probability at least $1 - \delta$, after $\mathcal{O}\left(\max_{1 \leq i \leq M-1} \frac{1}{(\Delta'_{(i)})^2} \log \frac{M}{\Delta'_{(i)}\delta}\right)$ calls for BQS with budget $\frac{3}{2} q_M$, the algorithm PLPAC terminates and outputs an $\epsilon$-optimal arm. Therefore, the total number of samples is $\mathcal{O}\left((M \log M) \max_{1 \leq i \leq M-1} \frac{1}{(\Delta'_{(i)})^2} \log \frac{M}{\Delta'_{(i)}\delta}\right)$.*

**Remark 5.** *The RankCentrality algorithm proposed in [23] converts the empirical pairwise marginals $\widehat{\mathbf{P}}$ into a row-stochastic matrix $\widehat{\mathbf{Q}}$. Then, considering $\widehat{\mathbf{Q}}$ as a transition matrix of a Markov chain, it ranks the items based on its stationary distribution. In [25], the authors show that if the pairwise marginals obey a PL distribution, this algorithm produces the mode of this distribution if the sample size is sufficiently large. In their setup, the learning algorithm has no influence on the selection of pairs to be compared; instead, comparisons are sampled using a fixed underlying distribution over the pairs. For any sampling distribution, their PAC bound is of order at least $M^3$, whereas our sample complexity bound in Theorem 4 is of order $M \log^2 M$.*

# 9  Experiments

Our approach strongly exploits the assumption of a data generating process that can be modeled by means of a PL distribution. The experimental studies presented in this section are mainly aimed at showing that it is doing so successfully, namely, that it has advantages compared to other approaches in situations where this model assumption is indeed valid. To this end, we work with synthetic data.

Nevertheless, in order to get an idea of the robustness of our algorithm toward violation of the model assumptions, some first experiments on real data are presented in Appendix I.[3]

## 9.1 The PAC-Item Problem

We compared our PLPAC algorithm with other preference-based algorithms applicable in our setting, namely INTERLEAVED FILTER (IF) [28], BEAT THE MEAN (BTM) [29] and MALLOWSMPI [7]. While each of these algorithms follows a successive elimination strategy and discards items one by one, they differ with regard to the sampling strategy they follow. Since the time horizon must be given in advance for IF, we run it with $T \in \{100, 1000, 10000\}$, subsequently referred to as IF($T$). The BTM algorithm can be accommodated into our setup as is (see Algorithm 3 in [29]). The MALLOWSMPI algorithm assumes a Mallows model [20] instead of PL as an underlying probability distribution over rankings, and it seeks to find the Condorcet winner—it can be applied in our setting, too, since a Condorcet winner does exist for PL. Since the baseline methods are not able to handle $\epsilon$-approximation except the BTM, we run our algorithm with $\epsilon = 0$ (and made sure that $v_i \neq v_j$ for all $1 \leq i \neq j \leq M$).

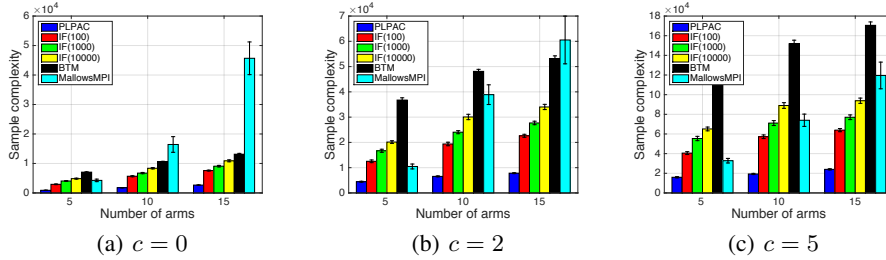

(a) $c = 0$      (b) $c = 2$      (c) $c = 5$

Figure 1: The sample complexity for $M = \{5, 10, 15\}$, $\delta = 0.1$, $\epsilon = 0$. The results are averaged over 100 repetitions.

We tested the learning algorithm by setting the parameters of PL to $v_i = 1/(c + i)$ with $c = \{0, 1, 2, 3, 5\}$. The parameter $c$ controls the complexity of the rank elicitation task, since the gaps between pairwise probabilities and $1/2$ are of the form $|p_{i,j} - 1/2| = |\frac{1}{2} - \frac{1}{1 + \frac{i+c}{j+c}}|$, which converges to zero as $c \to \infty$. We evaluated the algorithm on this test case with varying numbers of items $M = \{5, 10, 15\}$ and with various values of parameter $c$, and plotted the sample complexities, that is, the number of pairwise comparisons taken by the algorithms prior to termination. The results are shown in Figure 1 (only for $c = \{0, 2, 5\}$, the rest of the plots are deferred to Appendix E). As can be seen, the PLPAC algorithm significantly outperforms the baseline methods if the pairwise comparisons match with the model assumption, namely, they are drawn from the marginals of a PL distribution. MALLOWSMPI achieves a performance that is slightly worse than PLPAC for $M = 5$, and its performance is among the worst ones for $M = 15$. This can be explained by the elimination strategy of MALLOWSMPI, which heavily relies on the existence of a gap $\min_{i \neq j} |p_{i,j} - 1/2| > 0$ between all pairwise probabilities and $1/2$; in our test case, the minimal gap $p_{M,M-1} - 1/2 = \frac{1}{2 - 1/(c+M)} - 1/2 > 0$ is getting smaller with increasing $M$ and $c$. The poor performance of BTM for large $c$ and $M$ can be explained by the same argument.

## 9.2 The AMPR Problem

Since the RankCentrality algorithm produces the most probable ranking if the pairwise marginals obey a PL distribution and the sample size is sufficiently large (cf. Remark 5), it was taken as a baseline. Using the same test case as before, input data of various size was generated for RankCentrality based on uniform sampling of pairs to be compared. Its performance is shown by the black lines in Figure 2 (the results for $c = \{1, 3, 4\}$ are again deferred to Appendix F). The accuracy in a single run of the algorithm is 1 if the output of RankCentrality is identical with the most probable ranking, and 0 otherwise; this accuracy was averaged over 100 runs.

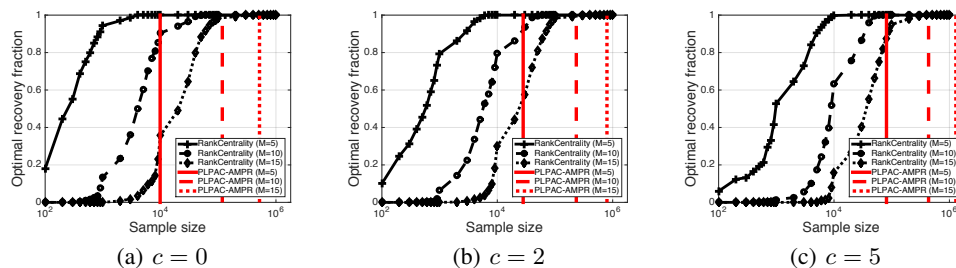

|  | (a) $c = 0$ | (b) $c = 2$ | (c) $c = 5$ |

Figure 2: Sample complexity for finding the approximately most probable ranking (AMPR) with parameters $M \in \{5, 10, 15\}$, $\delta = 0.05$, $\epsilon = 0$. The results are averaged over 100 repetitions.

We also run our PLPAC-AMPR algorithm and determined the number of pairwise comparisons it takes prior to termination. The horizontal lines in Figure 2 show the empirical sample complexity achieved by PLPAC-AMPR with $\epsilon = 0$. In accordance with Theorem 4, the accuracy of PLPAC-AMPR was always significantly higher than $1 - \delta$ (actually equal to 1 in almost every case).

As can be seen, RankCentrality slightly outperforms PLPAC-AMPR in terms of sample complexity, that is, it achieves an accuracy of 1 for a smaller number of pairwise comparisons. Keep in mind, however, that PLPAC-AMPR only terminates when its output is correct with probability at least $1 - \delta$. Moreover, it computes the confidence intervals for the statistics it uses based on the Chernoff-Hoeffding bound, which is known to be very conservative. As opposed to this, RankCentrality is an offline algorithm without any performance guarantee if the sample size in not sufficiently large (see Remark 5). Therefore, it is not surprising that, asymptotically, its empirical sample complexity shows a better behavior than the complexity of our online learner.

As a final remark, ranking distributions can principally be defined based on any sorting algorithm, for example MergeSort. However, to the best of our knowledge, pairwise stability has not yet been shown for any sorting algorithm other than QuickSort. We empirically tested the Merge-Sort algorithm in our experimental study, simply by using it in place of budgeted QuickSort in the PLPAC-AMPR algorithm. We found MergeSort inappropriate for the PL model, since the accuracy of PLPAC-AMPR, when being used with MergeSort instead of QuickSort, drastically drops on complex tasks; for details, see Appendix J. The question of pairwise stability of different sorting algorithms for various ranking distributions, such as the Mallows model, is an interesting research avenue to be explored.

## 10 Conclusion and Future Work

In this paper, we studied different problems of online rank elicitation based on pairwise comparisons under the assumption of a Plackett-Luce model. Taking advantage of this assumption, our idea is to construct a surrogate probability distribution over rankings based on a sorting procedure, namely QuickSort, for which the pairwise marginals provably coincide with the marginals of the PL distribution. In this way, we manage to exploit the (stochastic) transitivity properties of PL, which is at the origin of the efficiency of our approach, together with the idea of replacing the original Quick-Sort with a budgeted version of this algorithm. In addition to a formal performance and complexity analysis of our algorithms, we also presented first experimental studies showing the effectiveness of our approach.

Needless to say, in addition to the problems studied in this paper, there are many other interesting problems that can be tackled within the preference-based framework of online learning. For example, going beyond a single item or ranking, we may look for a good estimate $\widehat{\mathbb{P}}$ of the entire distribution $\mathbb{P}$, for example, an estimate with small Kullback-Leibler divergence: $\mathrm{KL}\left(\mathbb{P}, \widehat{\mathbb{P}}\right) < \epsilon$. With regard to the use of sorting algorithms, another interesting open question is the following: Is there any sorting algorithm with a worst case complexity of order $M \log M$, which preserves the marginal probabilities? This question might be difficult to answer since, as we conjecture, the MergeSort and the InsertionSort algorithms, which are both well-known algorithms with an $M \log M$ complexity, do not satisfy this property.

## Footnotes

[1]We omit the index $t$ if there is no danger of confusion.

[2]Put differently, $\tau$ is obtained from $\tau^B$ by continuing the execution of BQS ignoring the stopping criterion $B \leq 0$.

[3] In addition, we conducted some experiments to asses the impact of parameter $\epsilon$ and to test our algorithms based on Clopper-Pearson confidence intervals. These experiments are deferred to Appendix H and G due to lack of space.

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
