[Supplementary Material · PL_NIPS2015_final_5B_supp.pdf]

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

*Proof.* Consider the leaf of $T$ that is labeled by some set containing $i$ and $j$, and denote this set by $A'$. Now, consider running BQS on $A'$. By Theorem 1, the probability that it returns a ranking where $i$ precedes $j$ is exactly $\frac{v_i}{v_i + v_j}$. This implies the claim, because of the recursive nature of the process generating $\tau$ (i.e., by the property called "pairwise stability" by Ailon [1]).    □

Based on this observation, we can derive a result that guarantees the budgeting technique for the QuickSort algorithm to introduce no bias in the original marginal probabilities.

**Proposition 7** (Restatement of Proposition 2). *For any $B > 0$, any set $A \subseteq \mathcal{I}$ and any indices $i, j \in A$, the partial order $\mathbf{r} = \mathbf{r}_{\tau^B}$ generated by $\mathrm{BQS}(A, B)$ satisfies $\mathbb{P}(i \succ_{\mathbf{r}} j \,|\, \tau^B \in \mathcal{T}^B \setminus \mathcal{T}_{i,j}^B) = \frac{v_i}{v_i + v_j}$ .*

*Proof.* Denote by $\mathbf{r}'$ the ranking BQS would have returned for the budget $\infty$ instead of $B$. That is, $\mathbf{r}' = \mathbf{r}_\tau$, and so

$$\mathbb{P}(i \succ_{\mathbf{r}} j \,|\, \tau^B \in \mathcal{T}^B \setminus \mathcal{T}_{i,j}^B) = \mathbb{P}(i \succ_{\mathbf{r}'} j \,|\, \tau^B \in \mathcal{T}^B \setminus \mathcal{T}_{i,j}^B) \ . \tag{4}$$

Again, by Theorem 1,

$$\mathbb{P}(i \succ_{\mathbf{r}'} j) = \frac{v_i}{v_i + v_j} \ . \tag{5}$$

Additionally, by the previous Claim,

$$\mathbb{P}(i \succ_{\mathbf{r}'} j \,|\, \tau^B \in \mathcal{T}_{i,j}^B) = \frac{v_i}{v_i + v_j} \ . \tag{6}$$

Finally note that $\mathbb{P}(i \succ_{\mathbf{r}'} j) = \mathbb{P}(i \succ_{\mathbf{r}'} j \,|\, \tau^B \in \mathcal{T}^B \setminus \mathcal{T}_{i,j}^B) \cdot (1 - \mathbb{P}(\tau^B \in \mathcal{T}_{i,j}^B)) + \mathbb{P}(i \succ_{\mathbf{r}'} j \,|\, \tau^B \in \mathcal{T}_{i,j}^B) \cdot \mathbb{P}(\tau^B \in \mathcal{T}_{i,j}^B)$ and so

$$\mathbb{P}(i \succ_{\mathbf{r}'} j \,|\, \tau^B \in \mathcal{T}^B \setminus \mathcal{T}_{i,j}^B) = \frac{\mathbb{P}(i \succ_{\mathbf{r}'} j) - \mathbb{P}(i \succ_{\mathbf{r}'} j \,|\, \tau^B \in \mathcal{T}_{i,j}^B)\mathbb{P}(\tau^B \in \mathcal{T}_{i,j}^B)}{1 - \mathbb{P}(\tau^B \in \mathcal{T}_{i,j}^B)} \ . \tag{7}$$

The claim follows by putting together (4), (5), (6) and (7).    □

## B   Proof of Theorem 3

For the reader's convenience, we restate the theorem.

**Theorem 8.** *Set $\Delta_i = (1/2)\max\{\epsilon, p_{i^*,i} - 1/2\} = (1/2)\max\{\epsilon, \frac{v_{i^*} - v_i}{2(v_{i^*} + v_i)}\}$ for each index $i \neq i^*$. With probability at least $1 - \delta$, after $\mathcal{O}\left(\max_{i \neq i^*} \frac{1}{\Delta_i^2} \log \frac{M}{\Delta_i \delta}\right)$ calls for BQS with budget $M - 1$, PLPAC terminates and outputs an $\epsilon$-optimal arm. Therefore, the total number of samples is $\mathcal{O}\left(M \max_{i \neq i^*} \frac{1}{\Delta_i^2} \log \frac{M}{\Delta_i \delta}\right)$.*

*Proof.* Combining Proposition 2 with the Chernoff–Hoeffding bound implies that for any distinct indices $i$ and $j$ and for any $t \geq 1$, $\mathbb{P}\left(\widehat{p}_{i,j}^t \notin (p_{i,j} - c_{i,j}^t, p_{i,j} + c_{i,j}^t)\right) \leq \frac{\delta}{M^2 4 t^2}$, where

$$\widehat{p}_{i,j}^t = \frac{1}{n_{i,j}^t} \sum_{t'=1}^{t} \mathbb{I}\{i \succ_{\mathbf{r}^{t'}} j\}$$

and

$$n_{i,j}^t = \sum_{t'=1}^{t} \mathbb{I}\{i \succ_{\mathbf{r}^{t'}} j \text{ or } j \succ_{\mathbf{r}^{t'}} i\}$$

and $c_{i,j}^t = \sqrt{\frac{1}{n_{i,j}^t} \log \frac{8t^2 M^2}{\delta}}$. It thus holds that

$$\mathbb{P}\big((\forall i)(\forall j \neq i)(\forall t \geq 1)(\widehat{p}_{i,j}^t \in (p_{i,j} - c_{i,j}^t, p_{i,j} + c_{i,j}^t))\big) \geq 1 - \frac{\delta}{2} \ .$$

For the rest of the proof, assume that $(\forall i)(\forall j \neq i)(\forall t \geq 1)(\widehat{p}_{i,j}^t \in (p_{i,j} - c_{i,j}^t, p_{i,j} + c_{i,j}^t))$. This also implies correctness, and that if some arm is discarded, then it is indeed worse than some other arm. Consequently, arm $i^*$ will never be discarded, and thus $i^* \in A$ in each round.

Next, as it was pointed out, the BQS algorithm with budget $M-1$ results in a bucket order containing only two buckets since no recursive call is carried out with this budget. Thus the run of BQS simply consists of choosing a pivot item from $A$ uniformly at random and then dividing the rest of the items into two buckets based on comparing them to the pivot item. Let us denote the event of choosing an item $k$ for pivot element by $E_k$.

Now, consider some item $i \in A \setminus \{i^*\}$ which satisfies

$$\frac{|A|}{2} \leq \# \{k \in A : v_i \leq v_k\} \tag{8}$$

Then it holds that $i^*$ and $i$ end up in different buckets with probability

$$\begin{aligned}
\mathbb{P}\left(\tau \in \mathcal{T}^B \setminus \mathcal{T}_{i^*,i}^B\right) =& \mathbb{P}\left(i^* \succ_{\mathbf{r}} i \text{ or } i \succ_{\mathbf{r}} i^*\right) \\
\geq& \sum_{k \in A \setminus \{i^*,i\}} \mathbb{P}(E_k)\mathbb{P}(i^* \succ_{\mathbf{r}} i | E_k) \\
=& P(E_i)\mathbb{P}\left(i^* \succ_{\mathbf{r}} i\right) + P(E_{i^*})\mathbb{P}\left(i^* \succ_{\mathbf{r}} i\right) \\
& + \sum_{k \in A \setminus \{i^*,i\}} \mathbb{P}(E_k)\mathbb{P}\left(k \succ_{\mathbf{r}} i\right) \cdot \mathbb{P}\left(i^* \succ_{\mathbf{r}} k\right) \\
\geq& \frac{2}{|A|}p_{i^*,i} + \frac{1}{|A|} \sum_{k \in A \setminus \{i^*,i\}} p_{k,i}p_{i^*,k} \\
\geq& \frac{2}{|A|}p_{i^*,i} + \frac{\# \{k \in A \setminus \{i,i^*\} : v_i \leq v_k\}}{|A|}(1/2) \min_{k \neq i^*} p_{i^*,k} \\
\geq& \frac{\# \{k \in A : v_i \leq v_k\}}{2|A|} \min_{k \neq i^*} p_{i^*,k} \\
\geq& \frac{1}{4} \min_{k \neq i^*} p_{i^*,k} \geq \frac{1}{8} \tag{9}
\end{aligned}$$

where the last inequality follows from (8).

For every $t \geq 1$, denote by $F_t$ the set of arms $i$ which satisfies (8) in round $t$. Now, consider some subsequent rounds $t', t' + 1, \ldots, t''$, and some arm $i \in F_{t'}$. Let now $m_{i^*,i}^t = \sum_{\ell=t'}^{t} \mathbb{I}\{i^* \succ_{\mathbf{r}\ell} i \text{ or } i \succ_{\mathbf{r}\ell} i^* \text{ or } i \notin F_\ell \text{ or } i \notin A_\ell\}$. By (9), $\mathbb{E}[\mathbb{I}\{i^* \succ_{\mathbf{r}\ell} i \text{ or } i \succ_{\mathbf{r}\ell} i^* \text{ or } i \notin F_\ell \text{ or } i \notin A_\ell\}] \geq 1/8$ for any $t' \leq t \leq t''$ thus, according to the Chernoff-Hoeffding bound

$$\mathbb{P}\left(m_{i^*,i}^{t''} \leq \frac{t''-t'}{8} - \sqrt{(t''-t') \log \frac{2M}{\delta}}\right) \leq \frac{\delta}{2M} \ .$$

Consequently, $m_{i^*,i}^{t''} \geq \frac{1}{\Delta_i^2} \log \frac{8(t'')^2 M^2}{\delta}$ with probability at least $(1 - \delta/(2M))$ when

$$t'' - t' \geq \frac{16}{\Delta_i^2} \log \frac{8(t'')^2 M^2}{\delta} \ .$$

Recalling our assumption, it follows that arm $i$ gets discarded or $i \notin \cap_{t=t'}^{t''} F_t$, unless $p_{i^*,i} \leq 1/2+\epsilon$.

Also note that $\max_{t=t'}^{t''} |(A_t \cap F_{t'}) \setminus F_t| > 0$ means that at least $\max_{t=t'}^{t''} |(A_t \cap F_{t'}) \setminus F_t|$ arms in $A_{t'} \setminus F_{t'}$ got discarded between rounds $t'$ and $t''$. Defining $t_m = \frac{32m}{\Delta^2} \log \frac{8m \log M}{\delta}$ for $m = 1, 2, \ldots$, where $\Delta := \min_{i \neq i^*} \Delta_i$, it holds that

$$t_{m+1} - t_m \geq \frac{16}{\Delta^2} \log \frac{8(t_{m+1})^2 M^2}{\delta} \ ,$$

and thus the size of $A$ gets halved between $t_m$ and $t_{m+1}$ whenever $F_{t_m}$ only contains arms $i$ with $\Delta \geq \epsilon$. Noting that for any $t$ and for any $i \in F_t$ and $j \in A_t \setminus F_t$ it holds that $p_{j,i} \geq 1/2$, it follows that this halving continues as long as $A_{t_{m+1}}$ contains some arm $i$ with $p_{i^*,i} > 1/2 + \epsilon$. Accordingly, with probability at least $1 - \delta/2$, every arm $i$ with $p_{i^*,i} \geq (1/2) + \epsilon$ gets discarded after at most

$$\sum_{m=1}^{\lceil \log M \rceil} \frac{M}{2^m}(t_{m+1} - t_m) = \mathcal{O}\left(\sum_{m=1}^{\lceil \log M \rceil} \frac{M}{2^m} \frac{1}{\Delta^2} \log \frac{M}{\Delta \delta}\right) = \mathcal{O}\left(M \frac{1}{\Delta^2} \log \frac{M}{\delta}\right)$$

samples.

Finally, similarly as in (9), one can show that, if $p_{i^*,i} \leq 1/2 + \epsilon$ for every $i \in A$, then $\mathbb{P}(i^* \prec_{\mathbf{r}} i$ or $i \prec_{\mathbf{r}} i^*) \geq 1/4$. Therefore, with probability at least $1 - \delta \frac{|A|}{2M}$, after at most $\mathcal{O}\left(\frac{1}{\Delta^2} \log \frac{M}{\Delta \delta}\right)$ rounds, all pairs are compared at least $\mathcal{O}\left(\frac{1}{\Delta^2} \log \frac{M}{\Delta \delta}\right)$ times. This implies that after these rounds each confidence bounds gets small enough, and thus the termination criterion in line 13 of Algorithm 2 is satisfied. $\qquad\square$

## C  Pseudo-code of PLPAC-AMPR algorithm

---

**Algorithm 3** PLPAC-AMPR$(\delta, \epsilon)$

---

1: **for** $i = 1 \to M$ **do** $\qquad\qquad\qquad\qquad\qquad\qquad\qquad\qquad\qquad\qquad$ ▷ Initialization
2: $\qquad \underline{b}_i = 0$ and $\bar{b}_i = M$
3: $\qquad$ **for** $j = 1 \to M$ **do**
4: $\qquad\qquad \widehat{p}_{i,j} = 0 \qquad\qquad\qquad\qquad\qquad\qquad\qquad\qquad$ ▷ $\widehat{\mathbf{P}} = [\widehat{p}_{i,j}]_{M \times M}$
5: $\qquad\qquad n_{i,j} = 0 \qquad\qquad\qquad\qquad\qquad\qquad\qquad\qquad\quad$ ▷ $\widehat{\mathbf{N}} = [n_{i,j}]_{M \times M}$
6: set $A = \{1, \dots, M\}$
7: **repeat**
8: $\qquad$ compute $G = ([M], E)$ where $E = \{(i,j) \in [M]^2 : [\underline{b}_i, \bar{b}_i] \cap [\underline{b}_j, \bar{b}_j] \neq \emptyset\}$
9: $\qquad$ find the connected components $C_1, \dots, C_k$ of $G$
10: $\qquad$ **for** $i = 0 \to k$ **do**
11: $\qquad\qquad$ **if** $\#C_i > 1$ **then**
12: $\qquad\qquad\qquad q = 3(c_i + 1) \log c_i$ where $c_i = \#C_i$
13: $\qquad\qquad\qquad \mathbf{r} = \text{BQS}(C_i, q) \qquad\qquad\qquad\qquad$ ▷ Sorting based on PL model
14: $\qquad\qquad\qquad$ update the entries of $\widehat{\mathbf{P}}$ and $\mathbf{N}$ corresponding to $C_i$ based on $\mathbf{r}$
15: $\qquad$ set $c_{i,j} = \sqrt{\frac{1}{2n_{i,j}} \log \frac{4M^2 n_{i,j}^2}{\delta}}$ for all $i \neq j$
16: $\qquad$ **for** $i = 1 \to M$ **do**
17: $\qquad\qquad \underline{b}_i = \#\{j \in [M] \setminus \{i\} : \widehat{p}_{i,j} - c_{i,j} > 1/2\}$
18: $\qquad\qquad \bar{b}_i = \underline{b}_i + \#\{j \in [M] \setminus \{i\} : 1/2 \in [\widehat{p}_{i,j} - c_{i,j}, \widehat{p}_{i,j} + c_{i,j}]\}$
19: **until** $(\forall (i,j) \in [M]^2 : ((i \neq j) \wedge ([\underline{b}_i, \bar{b}_i] \cap [\underline{b}_j, \bar{b}_j] \neq \emptyset)) \to$
20: $\qquad\qquad\qquad\qquad ((1/2 - \epsilon < \widehat{p}_{i,j} - c_{i,j}) \wedge (1/2 + \epsilon > \widehat{p}_{i,j} + c_{i,j})))$
21: **return** $\text{argsort}(\underline{b}_1, \dots, \underline{b}_M) \qquad\qquad\qquad\qquad\qquad\qquad$ ▷ Break the ties arbitrarily

---

## D  Proof of Theorem 4

For the reader's convenience, we restate the theorem.

**Theorem 9.** *Set* $\Delta'_{(i)} = (1/2) \max\{\epsilon, \frac{v_{(i+1)} - v_{(i)}}{2(v_{(i+1)} + v_{(i)})}\}$ *for each* $1 \leq i \leq M$*, where* $v_{(i)}$ *denotes the i-th largest skill parameter. With probability at least* $1 - \delta$*, after* $\mathcal{O}\left(\max_{1 \leq i \leq M-1} \frac{1}{(\Delta'_{(i)})^2} \log \frac{M}{\Delta'_{(i)} \delta}\right)$ *calls for* BQS *with budget* $\frac{3}{2} q_M$*, the algorithm* PLPAC *terminates and outputs an* $\epsilon$*-optimal arm. Therefore, the total number of samples is* $\mathcal{O}\left((M \log M) \max_{1 \leq i \leq M-1} \frac{1}{(\Delta'_{(i)})^2} \log \frac{M}{\Delta'_{(i)} \delta}\right)$*.*

*Proof.* First we show that the confidence intervals contain the true parameters with high probability. Combining Proposition 2 with the Chernoff–Hoeffding bound implies that for any distinct indices $i$ and $j$ and for any $t \geq 1$, $\mathbb{P}\left(\widehat{p}_{i,j}^t \notin (p_{i,j} - c_{i,j}^t, p_{i,j} + c_{i,j}^t)\right) \leq \frac{\delta}{M^2 8 t^2}$, where $\widehat{p}_{i,j}^t = \frac{1}{n_{i,j}^t} \sum_{t'=1}^t \mathbb{I}\{i \succ_{\mathbf{r}^{t'}} j\}$, $n_{i,j}^t = \sum_{t'=1}^t \mathbb{I}\{i \succ_{\mathbf{r}^{t'}} j \text{ or } j \succ_{\mathbf{r}^{t'}} i\}$ and $c_{i,j}^t = \sqrt{\frac{1}{n_{i,j}^t} \log \frac{8 t^2 M^2}{\delta}}$. It thus holds that

$$\mathbb{P}\big((\forall i)(\forall j \neq i)(\forall t \geq 1)(\widehat{p}_{i,j}^t \in (p_{i,j} - c_{i,j}^t, p_{i,j} + c_{i,j}^t))\big) \geq 1 - \tfrac{\delta}{2} \ .$$

For the rest of the proof, assume that $(\forall i)(\forall j \neq i)(\forall t \geq 1)(\widehat{p}_{i,j}^t \in (p_{i,j} - c_{i,j}^t, p_{i,j} + c_{i,j}^t))$.

Now, we are going to show that when the algorithm terminates, it outputs an approximately most probable ranking (AMPR). The Copeland score of item $i$ is defined as

$$b_i = \#\{1 \leq j \leq M \,|\, (i \neq j) \wedge (p_{i,j} > 1/2)\} \ .$$

Moreover, we have

$$\underline{b} \leq b_i \leq \overline{b}_i$$

where

$$\underline{b}_i = \#\{j \in [M] \setminus \{i\} \,|\, \widehat{p}_{i,j} - c > 1/2\}$$

and $\overline{b}_i = \underline{b}_i + s_i$, where

$$s_i = \#\{j \in [M] \setminus \{i\} \,|\, 1/2 \in [\widehat{p}_{i,j} - c, \widehat{p}_{i,j} + c]\} \ .$$

Moreover it is easy to see that if $b_i > b_j$ for a pair of items $i \neq j$ then $v_i > v_j$. Therefore the ranking based on the Copeland score coincides with the ranking based on the skill parameters. Furthermore, according to the condition in line 20 of Algorithm 3, the algorithm does not terminate until, for any pair of items $i \neq j$ at least one of $[\underline{b}_i, \overline{b}_i] \cap [\underline{b}_j, \overline{b}_j] = \emptyset$ or $(1/2 - \epsilon < \widehat{p}_{i,j} - c_{i,j}) \wedge (1/2 + \epsilon > \widehat{p}_{i,j} + c_{i,j})$ holds. (The former implies that the pairwise order of $i$ and $j$ with respect to Copeland ranking is revealed, and the latter that $|p_{i,j} - 1/2| < \epsilon$.) This implies correctness.

In order to compute the sample complexity note that if for all indices $i \neq j$ and $k < k'$ such that $v_{(k)} = v_i$ and $v_{(k')} = v_j$, it holds that $n_{i,j}^t \geq \frac{1}{(\Delta_{(i)}')^2} \log \frac{8 t^2 M^2}{\delta}$ then, according to our assumption, $|\widehat{p}_{i,j}^t - p_{i,j}| \leq c_{i,j}^t \leq \Delta_i$. This implies that $[\underline{b}_i, \overline{b}_i] \cap [\underline{b}_j, \overline{b}_j] = \emptyset$ or $((1/2 - \epsilon < \widehat{p}_{i,j} - c_{i,j}) \wedge (1/2 + \epsilon > \widehat{p}_{i,j} + c_{i,j}))$. The algorithm thus terminates.

As the last step, we show that $n_{i,j}^t = \Omega\left(t/3 - \sqrt{t \log \frac{1}{\delta}}\right)$ with high probability, which then immediately implies the desired bound. First note that the running time of the (unstopped) QuickSort algorithm strongly concentrates around its expected value [22]. More precisely, with probability at least $1/2$, it uses at most $3M \log M + \mathcal{O}(\log M)$ comparisons to order a list of $M$ elements, and thus terminates without being stopped. Consequently, according to the Chernoff-Hoeffding bound, with probability at least $(1 - \frac{\delta}{8 t^2})$, BQS was stopped at most $t/2 + \sqrt{t \log \frac{8 t^2}{\delta}}$ times during the first $t$ subsequent runs. It thus holds with probability at least $1 - \delta/2$ that, for each $t \geq 1$, BQS was stopped at most $t/2 + \sqrt{t \log \frac{8 t^2}{\delta}}$ times during the first $t$ subsequent runs. This implies our claim, and thereby completes the proof of the sample complexity bound. $\qquad\square$

**Remark 10.** *The sample complexity analysis of* PLPAC-AMPR *does not take into account the acceleration step based on connected components implemented in line 9. Obviously this step does not affect the correctness of the algorithm, but might lead to a sample complexity bound which is lower than the one computed in Theorem 4. Nevertheless we leave to future work the analysis of how this step affects the sample complexity bound of* PLPAC-AMPR.

## E    The PAC-Item Problem

Figure 3: The sample complexity for $M = \{5, 10, 15\}$, $\delta = 0.1$, $\epsilon = 0$. The results are averaged over 100 repetitions.

# F The AMPR Problem

Figure 4: Sample complexity for finding the approximately most probable ranking (AMPR) with parameters $M \in \{5, 10, 15\}$, $\delta = 0.1$, $\epsilon = 0$. The results are averaged over 100 repetitions.

## G  Hoeffding vs. Clopper-Pearson

Our PLPAC and PLPAC-AMPR algorithms were devised based on the Hoeffding bound [16] which is known not to be tight in many cases. In our online learning framework, the learning algorithms make their decisions based on the estimates of binomial proportions which represent the pairwise probabilities of wins between pairs of items. Obviously, one might instantiate our algorithms with an exact confidence interval for binomial proportions such as Clopper-Pearson (CP) interval [10] instead of the one based on Hoeffding bound.

The Clopper-Pearson confidence interval can be written in the form of

$$\left[ B\left( \frac{\delta}{2}; b, n-b+1 \right), B\left( 1 - \frac{\delta}{2}; b+1, n-b \right) \right]$$

for a binomial sample consisting of $n$ observations with $b$ success events, and where $B(\tau; p, n)$ is the $\tau$th quantile from a beta distribution with shape parameters $p$ and $n$. We applied the same correction for the confidence parameter $\delta$ like in the case of Hoeffding bound, that is, the confidence parameter was set to $\delta/(t^2 M^2)$ in time step $t$.

In our first experiment, we tested our PLPAC along with CP confidence interval. As in Subsection 9.1, we set the parameters of underlying PL to $v_i = 1/(c+i)$ with $c = \{0, 1, 2, 3, 5\}$. We run an instance of our algorithm based on Hoeffding bound and one based on Clopper-Pearson confidence interval and compared their empirical sample complexity. The results are shown in Figure 5 for various number of arms. As one might expect, the PLPAC algorithm achieves a lower empirical sample complexity by using the CP interval. This is not striking since the the CP interval is usually tighter for binomial proportions than the one based on Hoeffding bound which is applicable for a more general class of random variables.

Figure 5: Sample complexity for finding the PAC-Item by using CP confidence interval. The parameters are $M \in \{5, 10, 15\}$, $\delta = 0.1$, $\epsilon = 0$. The results are averaged over 100 repetitions.

In our next experiment, we run the PLPAC-AMPR based on CP confidence interval. The results are shown in Figure 6. As before, the instance of PLPAC-AMPR based on Clopper-Pearson confidence interval achieve a lower empirical sample complexity than the one based on Hoeffding bound, however the relative improvement is not so pronounced like in the PAC-Item problem.

## H  Scaling with $\epsilon$

The goal of this experiment is to assess the impact of the parameters $\epsilon$ on the sample complexity. As in Subsection 9.1, we test the learning algorithm by setting the parameters of underlying PL to $v_i = 1/(c+i)$ with $c = \{0, 1, 2, 3, 5\}$ and run the PLPAC with various values for parameter $\epsilon$. The sample complexities are shown in Figure 7. As can be seen, the smaller $\epsilon$, the higher the sample complexity – thus, our algorithm scales gracefully with the approximation error allowed. This finding is especially important when the skill parameters of close-to-optimal items are very close to each other, and thus enormous number of pairwise comparisons are required to find the exact solution, i.e. item $i^*$.

(a) $c = 0$  (b) $c = 1$  (c) $c = 2$

(d) $c = 3$  (e) $c = 4$  (f) $c = 5$

Figure 6: Sample complexity for finding the approximately most probable ranking (AMPR) by using Clopper-Pearson confidence interval with parameters $M \in \{5, 10, 15\}$, $\delta = 0.1$, $\epsilon = 0$. The results are averaged over 100 repetitions. The performance of the RankCentrality is indicated by light grey lines in a similar way like in Figure 2 and 4.

(a) $M = 5$  (b) $M = 10$  (c) $M = 15$

Figure 7: Sample complexity for finding the PAC-Item with parameters $M \in \{5, 10, 15\}$, $\delta = 0.05$, $\epsilon \in \{0, 0.05, 0.1, 0.15, 0.2, 0.25, 0.3\}$. The results are averaged over 100 repetitions.

Next, we run the PLPAC-AMPR algorithm on the same task like above. The sample complexities are shown in Figure 8 for various values of parameter $c$. As can be seen, the smaller $\epsilon$, the higher the sample complexity as before justifying that our algorithm scales with parameter $\epsilon$ gracefully in this case as well.

## I  Results on real data

We conducted experiments on real data to asses the efficiency of our PLPAC method if the model assumption is violated to some extent. We used various datasets consisting of pairwise comparisons taken from the PrefLib ranking data repository[4]. The most important statistics of the datasets used in our experiment are shown in Table 1. To asses to what extent the data fit to a PL model, we calculated a statistic based goodness-of-fit statistic as follows. We compute an estimate for all $p_{i,j}$ based on the data which we denote $\tilde{p}_{i,j}$. Then we fit a model by using maximum likelihood (ML) estimator. Based on this fitted model, we computed the $p_{i,j}$ values. The statistic we calculated then is $\chi^2 = \sum_{i \neq j} (p_{i,j} - \tilde{p}_{i,j})^2 / p_{i,j}$. This statistic reflects to how the pairwise marginal probabilities

Figure 8: Sample complexity for finding the approximately most probable ranking (AMPR) with parameters $M \in \{5, 10, 15\}$, $\delta = 0.1$, $\epsilon \in \{0, 0.05, 0.1, 0.15, 0.2, 0.25\}$. The results are averaged over 100 repetitions.

computed based on the data fits to the ones that are computed based on the fitted model. Clearly, the more close $\chi^2$ is to 1, the more the data fits to a PL model (at least in terms of pairwise marginals).

| Data set | ID | Items | Rankings | $\chi^2$ |
|---|---|---|---|---|
| 1 | ED-00004-00000087 | 3 | 1662 | 0.13 |
| 2 | ED-00007-00000044 | 9 | 6990 | 1.31 |
| 3 | ED-00002-00000002 | 5 | 3742 | 9.31 |

Table 1: The most important statistics of the ranking datasets.

We run the PLPAC, MALLOWSMPI, BTM, and IF($T$) with $T \in \{100, 1000, 10000\}$ on the datasets listed in Table 1. The comparisons between two objects $i$ and $j$ are randomly drawn according to the empirical marginals. More detailed, the comparison of objects $i$ and $j$ are generated from **Bern**$(\tilde{p}_{ij})$. We plotted the average of their sample complexity over 100 repetitions which is shown in Figure 9.

Figure 9: Sample complexity for finding the PAC-Item by using benchmark datasets.

The results reveal a few general trends. First, the improvement of PLPAC in terms of sample complexity is more pronounced on the datasets which meets more with our model assumption,

namely, it fits better to PL model. Note that, the better the fit, the more close $\chi^2$ value is to $1$. Second, the MALLOWSMPI learning algorithm requires more samples than the other learning algorithms on the "ED-00007-00000044" data. This might be explained by the fact that its elimination strategy consists of selecting two random items and comparing them until one beats the other, i.e. the estimate for the pairwise probabilities significantly differs from $1/2$. Since the empirical pairwise marginals for some items are close to $1/2$ for this dataset, if the MALLOWSMPI algorithm selects such a pair of items, then it compares them many times until one of them gets eliminated. In addition, this argument also explains the higher variance of sample complexity in this case.

## J Inapplicability of Merge Sort algorithm for sampling Plackett-Luce model

In principle, ranking distributions can be defined based on any other sorting algorithm, too, for example MergeSort. However, to the best of our knowledge, pairwise stability with PL distribution has not yet been shown for any sorting algorithm other than QuickSort. We empirically tested the MergeSort algorithm, simply by using it in place of the budgeted QuickSort. We run the PLPAC-AMPR algorithm on the test case we used in our experimental study earlier, that is, the parameters of the PL is set to $v_i = 1/(c+i)$. We set the $M = 5$ and $c = \{0, 1, 2, 3, 4, 5\}$. The sample complexity and the accuracy of PLPAC-AMPR based on Merge Sort algorithm with respect to parameter $c$ is shown in Figure 10. Based on the plots, this version of PLPAC-AMPR is also efficient in terms of sample complexity, but the accuracy of the algorithms drops for hard learning task when $c > 3$. This might be explained by the fact that the Merge Sort algorithm does not have the property of pairwise stability for PL distributions. Yet, we could not show this claim formally so far.

(a) Sample complexity          (b) Accuracy

Figure 10: Sample complexity and accuracy of PLPAC-AMPR along with Merge Sort based sampling for finding the approximately most probable ranking (AMPR) with parameters $M = 5$, $\delta = 0.1$, $\epsilon = 0$. The results are averaged over $100$ repetitions.