[Reviews · NeurIPS 2015]

Submitted by Assigned_Reviewer_1

Small comments:

- It should be mentioned explicitly somewhere that the pairwise marginals of the PL model are just a BTL distribution

- Rem 5: the bound in [25] is for exact recovery, that given here for approximate recovery; this should be made clear

- sec 9.1: several places: c or m [= --> \in] {range}

- p.7, last line: horizontal --> vertical

- sec 9.2: another interesting experiment would be to compare with rank centrality in terms of not optimal recovery fraction, but say fraction of times an \epsilon-optimal ranking is recovered (for some \epsilon > 0); does your algorithm give better sample complexity in that case?

Summary: The paper considers the problem of identifying (approximately) the most probable ranking over n items - or just the top-ranked item - assuming an underlying Plackett-Luce (PL) distribution, from active queries of pairwise comparisons from the distribution (this type of problem is also known as a dueling bandits problem). It proposes algorithms for both settings (identifying full ranking and identifying top-ranked item) based on a budgeted quicksort procedure that is shown to preserve pairwise comparison probabilities under the PL distribution. The paper gives query complexity bounds for both settings. Experiments show that under the PL model, the algorithm for finding the top item outperforms several other recently proposed algorithms. On the other hand, the algorithm for finding a full ranking does not outperform a simple passive sampling algorithm that uses a naive uniform sampling strategy, leaving open the possibility of designing a better active algorithm in this case.

The paper is exceptionally well written and has interesting results and very well-designed experiments.

Submitted by Assigned_Reviewer_2

In the setting of dueling bandits and under the PL model assumption, the authors consider two tasks: finding an approximate best arm and finding an approximate ranking of the arms. Both tasks relates to the estimations of the pairwise marginals of the underlying PL model, which is done with an early-stopped version of the QuickSort. This Budgeted QuickSort allows to build algorithms which sample complexity is O(M log^2 M). Experiments compare the sample complexity of the proposed algorithms against state-of-art methods.

The problem is not well-motivated, which is even more visible by the lack of experiments based on real-world dataset. This been said, I like the algorithms for their simplicity and the theoretical study that support them is correct. I didn't check proofs in appendix, but the results seems realistic. I would have appreciated a little discussion about the choice of the method to build confidence intervals in PLPAC -- e.g. is the algorithm sensitive to the method ?

The paper is correctly organized and reasonably easy to follow. I still find a bit curious to put an algorithm (AMPR-PLPAC) in supplementary material.

As I was saying, the problem is not well-motivated but the gain in terms of sample conplexity over existing algorithms is significant.
Summary: The algorithm is interesting, the theoretical study is complete. However, problem is not well-motivated and looks like a niche problem. This feeling is strengthen by the experiments that are only on synthetic data.

Submitted by Assigned_Reviewer_3

This paper proposes the use of a (Budgeted) Quick-Sort algorithm for the rank elicitation problem (in the dueling bandits setting). The algorithm works by fitting the observations to those observed under a Plackett-Luce distribution. Under this assumption the model is then analyzed theoretically and studied empirically.

While I like the overall idea in the paper, I had a few concerns starting with the restrictiveness of the model. Compared to closely related work, such as the vast work on the Dueling Bandits, the model here is analyzed under a very strict assumption of having been generated under a P-L model. Given that both the theoretical and empirical analysis is under this assumption, I'm unclear how impactful this will be given other existing algorithm. Some analysis and experimentation with even the smallest amount of violation from this P-L assumption would have added significant value in my opinion (given that real world data has no guarantees of coming from a P-L model). I also would have preferred ranking metrics to study performance at the AMPR problem in addition to optimal recovery rate, to better understand where the errors lie.
Summary: Interesting application of QuickSort to the dueling bandits problem. However unsure of impact due to limiting model assumptions.

Submitted by Assigned_Reviewer_4

The authors propose an elicitation algorithm to compute the ranking over alternatives w.r.t. their parameters in Placket-Luce. The elicitation is guided quick sort and the sample complexity is bounded.

The paper is very well written. It clearly made a solid contribution to a natural and important problem. It would be nicer to see an lower bound on the sample complexity in the PAC framework for the problem considered in the paper. The main reason for not giving a higher recommendation is the presence of [1], which seems to have most of the conceptual setup of the model and problem.
Summary: This is a clearly-written paper that made a solid (but not earthshaking) contribution to elicitation under the Plackett-Luce model.

Submitted by Assigned_Reviewer_5

In this paper, the authors present dueling bandit approach where the arms are assumed to be ranked according to Placket-Luce distribution.

Typically, bandit problems have regret analysis. What is presented in this paper is sample complexity: the number of samples required to obtain an \epsilon-optimal arm with probability 1 - \delta. I did not see any discussion about the regret bounds of the proposed algorithms. Also, how does the regret bound of the proposed algorithm compare with that of other existing

dueling bandit algorithms? It was not clear from reading the paper.

I think that one major limitation of the paper is that the experiments are based on synthetic data. It is not clear when the PL distribution assumption holds and to what problems, the proposed approach is applicable. The experiments seem too artificial.

Other comments:

Line 202: what is "end" after [M],?

Line 7, algorithm 2, N should be \hat{N}?

The authors present an algorithm in Supplementary material and its analysis in the main paper. I think it should be other way round.
Summary: Yet another approach for dueling bandit. Experiments are weak, it is not clear from the paper how the proposed method is better than existing approaches.

Submitted by Assigned_Reviewer_6

The authors have done a very good job in explaining the relevant literature in online preference based mechanisms. Both PAC-item and AMPR preference-based approximations are considered as goals for the problem of dueling bandit ranker. The authors build upon the fact that the pairwise comparisons executed by QuickSort algorithm in a stochastic setting are drawn from the pairwise marginals of the Plackett-Luce model. The idea of the Budgeted Quick-sort based algorithm seems simple enough -- however, I am not entirely convinced about its novelty. The elimination strategy used both for the PAC problem and the AMPR seems very intuitive (eliminate an item significantly beaten by another item). For the AMPR problem the authors estimate the Copeland score for every item. One contribution of the paper is that the authors give sample complexity bounds for both PAC and AMPR. The synthetic data results mostly follow the bounds covered in the theorems. One note is that the Condorcet winner is too restrictive of an assumption, thus "forcing" the authors to focus on the Placket Luce model which satisfies its existence.

Summary: The contribution of this paper is to use a budgeted version of Quicksort to construct surrogate probability distributions over rankings. Based on the fact that the pairwise marginals using Quicksort coincide with the marginals of the Plackett Luce mode, they are able to take advantage of the transitivity properties of Placket-Luce.

Author Feedback
Author rebuttal: We like to thank the reviewers for their positive feedback!

General comments:

- Although we agree that the assumption of the Plackett-Luce model (as a generalization of the Bradley-Terry model) may appear restrictive and will certainly not be satisfied in all practical applications, we like to emphasize that the PL model, in addition to the Mallows model, is the standard model in the statistics of rank data and widely used in many fields of applied statistics, e.g., voting and discrete choice theory in economics -- its status in these fields is comparable to the status of the Gaussian distribution for real-valued data. Therefore, we are convinced that studying the dueling bandits problem under this assumption is a worthwhile endeavor. In this regard, we also like to mention that the PL model has already been studied in the context of other preference learning problems as well (for example, see papers at ICML 2009 and 2010).

- Complying with the request of several reviewers, we are going to put the pseudo-code of the PLPAC-AMPR into the main paper.

Rev 1:

The confidence intervals in our paper are derived from Hoeffding's inequality in a standard way. Yet, other bounds could be used by our algorithm as well, for example bounds based on the Clopper-Pearson (CP) exact confidence interval (the pairwise comparisons are Bernoulli random variables). Since the CP interval is tighter, the empirical performance of our method would probably improve. The formal analysis of sample complexity becomes much harder for CP, however. Anyway, we are going to evaluate our algorithms by using CP interval and add some experiments and discussion to the supplementary.

Rev 2:

Regarding the regret bound: Our main focus in this paper is the PAC setup, not the regret-minimization setting. Nevertheless, as first pointed out by Yue et. al. (The K-armed dueling bandits problem, 2012), it is possible to compute a regret bound for each PAC algorithm that identifies the best arm. For a detailed discussion on this issue, see Urvoy et. al.: Generic Exploration and K-armed Voting Bandits, 2013, Appendix B.1. The technique is called ``explore-then-exploit'': The PAC algorithm tries to identify the best arm (Condorcet winner) with high probability in the first couple of rounds, and fully commits to the arm found to be best for the rest of the time (i.e., repeatedly compares this arm to itself). Since we assume the PL model, the Condorcet winner is guaranteed to exist and thus the regret notion of Yue at. al. is well-defined. Therefore, our method is amenable to the ``explore-then-exploit'' technique, allowing a regret bound to be computed for PLPAC. We are going to add a comment on this issue.

regarding line 202: ``end'' should be ``and''

Rev 3:

The first application of quicksort to ranking under PL model was in Ailon, NIPS, 2015, and to the best of our knowledge, ours is the first application where it is required to terminate after Mlog M comparisons with high probability---and thus a budgeted version is needed.

Rev 4:

Making experiments violating the model assumption is indeed a valid point. In particular, it would be interesting to analyze the robustness of our algorithm against deviations from the PL model. The only trouble with this type of experiment is that SOME model is obviously needed to generate the data, and any choice of a "non-PL model" may appear arbitrary to some extent. What does it mean that a model is not PL, and how to measure the degree of violation from PL?

Evaluating PLPAC-AMPR in terms of other ranking metrics is indeed important to better understand why a problem instance is hard from a learning point of view. We shall evaluate our algorithm in this way as well, and add some discussion to the supplementary material.

Rev 5:

Regarding the \epsilon-optimal ranking recovery: our algorithm achieves lower sample complexity if \epsilon>0 like the PAC bandit algorithms. We are going to evaluate the PLPAC-AMPR algorithm for \epsilon > 0, and add these results to Fig. 2.

Rev 6:

The remark on the lower bounds is absolutely valid. In specific cases, the lower bound techniques of the multi-armed bandit problem apply directly. The PL assumption assumes a utility-based representation of the arms, which makes our setup similar to the value-based setting to some extent. The usual approach (Mannor & Tsitsiklis, 2004) might work, and one might be able to show lower bounds that match (up to logarithmic factor) the upper bound for PLPAC-AMPR. This will be made explicit in the final version.

Regarding the relation of our work to [1], see our reply to Reviewer 3.